# EasyTPP: Towards Open Benchmarking the Temporal Point Processes

**Siqiao Xue**[1]   **Xiaoming Shi**[1]   **Zhixuan Chu**[1]   **Yan Wang**[1]   **Hongyan Hao**[1]
**Caigao Jiang**[1]   **Chen Pan**[1]   **Yi Xu**[1]   **James Y. Zhang**[1]
**Qingsong Wen**[2]   **Jun Zhou**[1]   **Hongyuan Mei**[3]
[1]Ant Group   [2]Alibaba DAMO Academy   [3]TTIC
{siqiao.xsq,peter.sxm,chuzhixuan.czx,luli.wy,hongyanhao.hhy}@antgroup.com
{caigao.jcg,bopu.pc,haolin.xy,james.z,jun.zhoujun}@antgroup.com
qingsong.wen@alibaba-inc.com   hongyuan@ttic.edu

## Abstract

Continuous-time event sequences play a vital role in real-world domains such as healthcare, finance, online shopping, social networks, and so on. To model such data, temporal point processes (TPPs) have emerged as the most natural and competitive models, making a significant impact in both academic and application communities. Despite the emergence of many powerful models in recent years, there hasn't been a central benchmark for these models and future research endeavors. This lack of standardization impedes researchers and practitioners from comparing methods and reproducing results, potentially slowing down progress in this field. In this paper, we present EasyTPP, the first central repository of research assets (e.g., data, models, evaluation programs, documentations) in the area of event sequence modeling. Our EasyTPP makes several unique contributions to this area: a unified interface of using existing datasets and adding new datasets; a wide range of evaluation programs that are easy to use and extend as well as facilitate reproducible research; implementations of popular neural TPPs, together with a rich library of modules by composing which one could quickly build complex models. Our benchmark is open-sourced: all the data and implementation can be found at this Github repository.[1] We will actively maintain this benchmark and welcome contributions from other researchers and practitioners. Our benchmark will help promote reproducible research in this field, thus accelerating research progress as well as making more significant real-world impacts.

## 1   Introduction

Continuous-time event sequences are ubiquitous in various real-world domains, such as neural spike trains in neuroscience (Williams et al., 2020), orders in financial transactions (Jin et al., 2020), and user page viewing behavior in the e-commerce platform (Hernandez et al., 2017). To model these event sequences, temporal point processes (TPPs) are commonly used, which specify the probability of each event type's instantaneous occurrence, also known as the *intensity function*, conditioned on the past event history. Classical TPPs, such as Poisson processes (Daley & Vere-Jones, 2007) and Hawkes processes (Hawkes, 1971), have a well-established mathematical foundation and have been widely used to model traffic (Cramér, 1969), finance (Hasbrouck, 1991) and seismology (Ogata, 1988) for several decades. However, the strong parametric assumptions in these models constrain

---

[1]https://github.com/ant-research/EasyTemporalPointProcess.

Submitted to the 37th Conference on Neural Information Processing Systems (NeurIPS 2023) Track on Datasets and Benchmarks. Do not distribute.

their ability to capture the complexity of real-world phenomena. To overcome the limitations of classical TPPs, many researchers have been developing neural versions of TPPs, which leverage the expressiveness of neural networks to learn complex dependencies; see section 7 for a comprehensive discussion. Since then, numerous advancements have been made in this field, as evidenced by the rapidly growing literature on neural TPPs since 2016. Recent reviews have documented the extensive methodological developments in TPPs, which have expanded their applicability to various real-world scenarios. As shown in Figure 2 and Appendix F.1, the number of research papers on TPPs has been steadily increasing, indicating the growing interest and potential impact of this research area. These advancements have enabled more accurate and flexible modeling of event sequences in diverse fields.

In this work, inspired by Hugging Face (Wolf et al., 2020) for computer vision and natural language processing, we take the initiative to build a central library, namely EasyTPP, of popular research assets (e.g., data, models, evaluation methods, documentations) with the following distinct merits:

1. **Standardization**. We establish a standardized benchmark to enable transparent comparison of models. Our benchmark currently hosts 5 popularly-used real-world datasets that cover diverse real-world domains (e.g., commercial, social), and will include datasets in other domains (e.g., earthquake and volcano eruptions). One of our contributions is to develop a unified format for these datasets and provide sourcre code (with thorough documentation) for data processing. This effort will free future researchers from large amounts of data-processing work, and facilitate exploration in new research topics such as transfer learning and adaptation (see Section 6).

2. **Comprehensiveness**. Our second contribution is to provide a wide range of easy-to-use evaluation programs, covering popular evaluation metrics (e.g., log-likelihood, kinds of next-event prediction accuracies and sequence similarities) and significance tests (e.g., permutation tests). By using this shared set of evaluation programs, researchers in this area will not only achieve a higher pace of development, but also ensure a better reproducibility of their results.

3. **Convenience**. Another contribution of EasyTPP is a rich suite of modules (functions and classes) which will significantly facilitate future method development. We reproduced previous (eight most-cited and competitive) models by composing these modules like building LEGOs; other researchers can reuse the modules to build their new models, significantly accelerating their implementation and improving their development experience. Examples of modules are presented in section 3.

4. **Flexibility**. Our library is compatible with both PyTorch (Paszke et al., 2019) and Tensor-Flow (Abadi et al., 2016), the top-2 popular deep learning frameworks, and thus offers a great flexibility for future research in method development.

5. **Extensibility**. Following our documentation and protocols, one could easily extend the EasyTPP library by adding new datasets, new modules, new models, and new evaluation programs. This high extensibility will contribute to building a healthy open-source community, eventually benefiting the research area of event sequence modeling.

## 2 Background

**Definition.** Suppose we are given a fixed time interval $[0, T]$ over which an event sequence is observed. Suppose there are $I$ events in the sequence at times $0 < t_1 < \ldots < t_I \le T$. We denote the sequence as $x_{[0,T]} = (t_1, k_1), \ldots, (t_I, k_I)$ where each $k_i \in \{1, \ldots, K\}$ is a discrete event type. Note that representations in terms of time $t_i$ and the corresponding inter-event time $\tau_i = t_i - t_{i-1}$ are isomorphic, we use them interchangeably. TPPs are probabilistic models for such event sequences. If we use $p_k(t \mid x_{[0,t)})$ to denote the probability that an event of type $k$ occurs over the in-

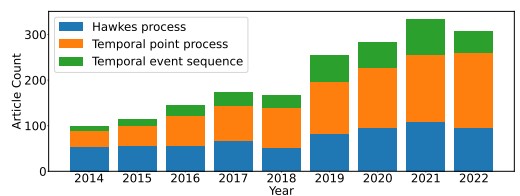

Figure 2: ArXiv submissions over time on TPPs

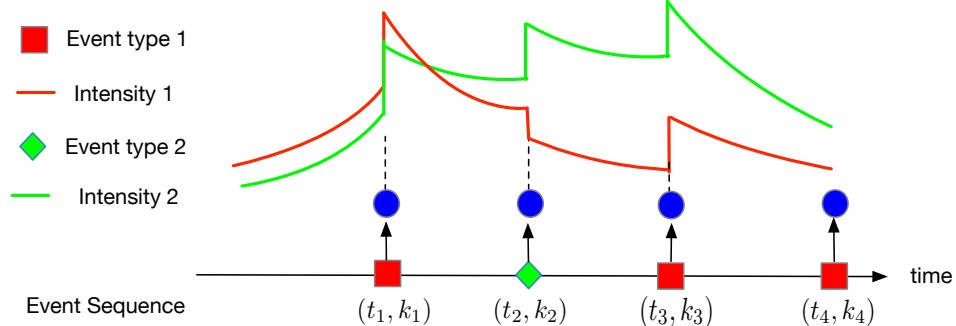

Figure 1: Drawing an event stream from a neural TPP. The model reads the sequence of past events (polygons) to arrive at a hidden state (blue). That state determines the future "intensities" of the two types of events–that is, their time-varying instantaneous probabilities. The intensity functions are continuous parametric curves (solid lines) determined by the most recent RNN state. In this example, events of type 1 excite type 1 but inhibit type 2. Type 2 excites itself and type 1. Those are immediate effects, shown by the sudden jumps in intensity.

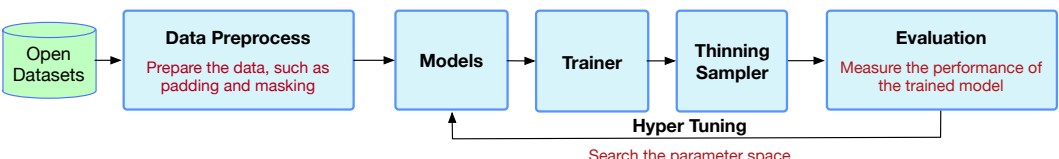

Figure 3: An open benchmarking pipeline using EasyTPP.

finitesimal interval $[t, t + dt)$, then the probability that nothing occurs will be $1 - \sum_{k=1}^{K} p_k(t \mid x_{[0,t)})$. Formally, the distribution of a TPP can be characterized by the **intensity** $\lambda_k(t \mid x_{[0,t)}) \geq 0$ for each event type $k$ at each time $t > 0$ such that $p_k(t \mid x_{[0,t)}) = \lambda_k(t \mid x_{[0,t)})dt$.

**Neural TPPs.** A neural TPP model autoregressively generates events one after another via neural networks. A schematic example is shown in Figure 1 and a detailed description on data samples can be found at our online documentation. For the $i$-th event $(t_i, k_i)$, it computes the embedding of the event $e_i \in \mathbb{R}^D$ via an embedding layer and the hidden state $h_i$ gets updated conditioned on $e_i$ and the previous state $h_{i-1}$. Then one can draw the next event conditioned on the hidden state $h_i$:

$$t_{i+1}, k_{i+1} \sim \mathbb{P}_\theta(t_{i+1}, k_{i+1} | h_i), \quad h_i = f_{update}(h_{i-1}, e_i), \tag{1}$$

where $f_{update}$ denotes a recurrent encoder, which could be either RNN (Du et al., 2016; Mei & Eisner, 2017) or more expressive attention-based recursion layer (Zhang et al., 2020; Zuo et al., 2020; Yang et al., 2022). A new line of research models the evolution of the states completely in continuous time:

$$h_{i-} = f_{evo}(h_{i-1}, t_{i-1}, t_i) \quad \text{between event times} \tag{2}$$

$$h_i = f_{update}(h_{i-}, e_i) \quad \text{at event time } t_i \tag{3}$$

The state evolution in Equation (2) is generally governed by an ordinary differential equation (ODE) (Rubanova et al., 2019). For a broad and fair comparison, in EasyTPP, we implement not only recurrent TPPs but also an ODE-based continuous-time state model.

**Learning TPPs.** Negative log-likelihood (NLL) is the default training objective for both classical and neural TPPs. The NLL of a TPP given the entire event sequence $x_{[0,T]}$ is

$$\sum_{i=1}^{I} \log \lambda_{k_i}(t_i \mid x_{[0,t_i)}) - \int_{t=0}^{T} \sum_{k=1}^{K} \lambda_k(t \mid x_{[0,t)})dt \tag{4}$$

Derivations of this formula can be found in previous work Hawkes (1971); Mei & Eisner (2017).

## 3 Benchmarking Process

Figure 3 presents the open benchmarking pipeline for neural TPPs, which is implemented in EasyTPP. In summary, the pipeline consists of the following key components.

**Data Preprocess.** Following common practices, we split the set of sequences into the train, validation, and test set with a fixed ratio. To feed the sequences of varying lengths into the model, in EasyTPP, we pad all sequences to the same length, then use the "sequence_mask" tensor to identify which event tokens are padding. As we implemented several variants of attention-based TPPs, we also generated the "attention_mask" to mask all the future positions at each event to avoid "peeking into the future".

**Model Implementation.** Our EasyTPP library provides a suite of modules, and one could easily build complex models by composing these modules. Specifically, we implemented the models (see section 5.1) evaluated in this paper with our suite of modules (e.g., continuous-time LSTM, continuous-time attention). Moreover, some modules are model-agnostic methods for training and inference, which will further speed up the development speed of future methodology research. Below are two signature examples:

- compute_loglikelihood (function), which calculates log-likelihood of a model given data. It is non-trivial to correctly implement it due to the integral term of log-likelihood in Equation (4), and we have found errors in popular implementations.

- EventSampler (class), which draws events from a given point process via the thinning algorithm. The thinning algorithm is commonly used in inference but it is non-trivial to implement (and rare to see) an efficient and batched version. Our efficient and batched version (which we took great efforts to implement) will be useful for nearly all intensity-based event sequence models.

**Training.** We can estimate the model parameters by locally maximizing the NLL in Equation (4) with any stochastic gradient method. Note that computing the NLL can be challenging due to the presence of the integral in the second term in Equation (4). In EasyTPP, by default, we approximate the integral by Monte-Carlo estimation to compute the overall NLL (see Appendix C.1). Nonetheless, EasyTPP also incorporates some neural TPPs (e.g., the intensity-free model (Shchur et al., 2020)), which allow us to compute the NLL analytically, which is more computationally efficient.

**Sampling.** Given the learned parameters, we apply the minimum Bayes risk (MBR) principle to predict the time and type with the lowest expected loss. A recipe can be found in Appendix C.2. Note that other methods exist for predicting a TPP, such as adding an MLP layer to directly output the time and type prediction (Zuo et al., 2020; Zhang et al., 2020). However, as we aim to build a generative model of event sequences, we believe the principal way to make predictions based on continuous-time generative model is thinning algorithm (Ogata, 1988). In EasyTPP, a batch-wise thinning algorithm is consistently used when evaluating the predictive performance of TPPs.

**Hyperparameter Tuning.** Most studies specified the detailed hyper-parameters of their models in the papers. However, with the modified code fitted in the EasyTPP framework or the new splits of datasets, it may be inappropriate to use the same hyper-parameters. Besides the classical grid search method, we also integrate *Optuna* (Akiba et al., 2019) in our framework to automatically search optimal hyperparameters and prune unpromising trials for faster results.

We hope that the definition of our open benchmarking pipeline could provide guidance for fair comparisons and reproducible works in TPPs.

## 4 `EasyTPP`'s Software Interface

**High Level Software Architecture.** The purpose of building EasyTPP is to provide a simple and standardized framework to allow users to apply different state-of-the-art (SOTA) TPPs to arbitrary data sets. For researchers, EasyTPP provides an implementation interface to integrate new recourse methods in an easy-to-use way, which allows them to compare their method to already existing methods. For industrial practitioners, the availability of benchmarking code helps them easily assess the applicability of TPP models for their own problems.

High-level visualization of the EasyTPP's software architecture is depicted in Figure 9. *Data Preprocess* component provides a common way to access the event data across the software and maintains information about the features. For the *Model* component, the library provides the possibility to use

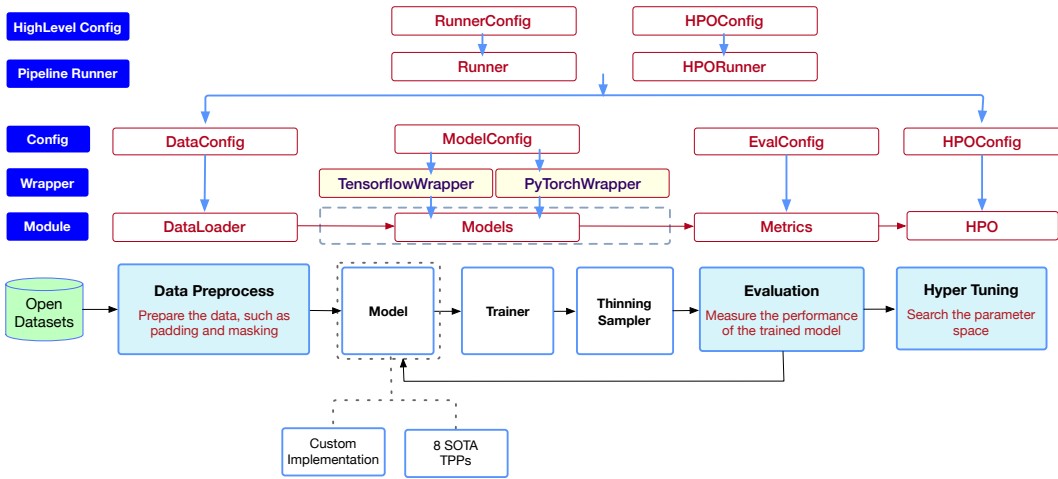

Figure 4: Architecture of the EasyTPP library. The dashed arrows show the different implementation possibilities, either to use pre-defined SOTA TPP models or provide a custom implementation. All dependencies between the configurations and modules are visualized by solid arrows with additional descriptions. Overall, the running of the pipeline is parameterized by the configuration classes - *RunnerConfig* (w/o hyper tuning) and *HPOConfig* (with hyper tuning).

existing methods or extend the users' custom methods and implementations. A *wrapper* encapsulates the black-box models along with the trainer and sampler. The primary purpose of the wrapper is to provide a common interface to easily fit in the training and evaluation pipeline, independently of their framework (e.g., PyTorch, TensorFlow). The running of the pipeline is parameterized by the configuration class - *RunnerConfig* (without hyper-parameter tuning) and *HPOConfig* (with hyper-parameter tuning).

**Why and How Does EasyTPP Support Both PyTorch and TensorFlow?** PyTorch and TensorFlow are the two most popular Deep Learning (DL) frameworks today. PyTorch has a reputation for being a research-focused framework, and indeed, most of the authors have implemented TPPs in PyTorch, which are used as references by EasyTPP. On the other hand, TensorFlow has been widely used in real world applications. For example, Microsoft recommender ,[2] NVIDIA Merlin [3] and Alibaba EasyRec [4] are well-known industrial user modeling systems with TensorFlow as the backend. In recent works, TPPs have been introduced to better capture the evolution of the user preference in continuous-time (Bao & Zhang, 2021; Fan et al., 2021; Bai et al., 2019). To support the use of TPPs by industrial practitioners, we implement an equivalent set of TPPs in TensorFlow. As a result, EasyTPP not only helps researchers analyze the strengths and bottlenecks of existing models, but also facilitates the deployment of TPPs in industrial applications.

See Appendix B for more details on the interface and examples of difference user cases.

## 5 Experimental Evaluation

### 5.1 Experimental Setup

We comprehensively evaluate 9 models in our benchmark, which include the classical **Multivariate HakwesProcess(MHP)** and 8 widely-cited state-of-the-art neural models:

- Two RNN-based models: **Recurrent marked temporal point process (RMTPP)** (Du et al., 2016) and **neural Hawkes Process (NHP)** (Mei & Eisner, 2017).

---

[2]https://github.com/microsoft/recommenders.
[3]https://developer.nvidia.com/nvidia-merlin.
[4]https://github.com/alibaba/EasyRec.

- Three attention-based models: **self-attentive Hawkes pocess (SAHP)** (Zhang et al., 2020), **transformer Hawkes process (THP)** (Zuo et al., 2020), **attentive neural Hawkes process (AttNHP)** (Yang et al., 2022).

- One TPP with the fully neural network based intensity: **FullyNN** (Omi et al., 2019).

- One intensity-free model **IFTPP** (Shchur et al., 2020).

- One TPP with the hidden state evolution governed by a neural ODE: **ODETPP**. It is a simplified version of the TPP proposed by Chen et al. (2021) by removing the spatial component. .

We conduct experiments on 1 synthetic and 5 real-world datasets from popular works that contain diverse characteristics in terms of their application domains and temporal statistics (see Table 2):

- **Synthetic**. This dataset contains synthetic event sequences from a univariate Hawkes process sampled using `Tick` (Bacry et al., 2017) whose conditional intensity function is defined by $\lambda(t) = \mu + \sum_{t_i < t} \alpha\beta \cdot \exp\left(-\beta(t - t_i)\right)$ with $\mu = 0.2, \alpha = 0.8, \beta = 1.0$. We randomly sampled disjoint train, dev, and test sets with 1200, 200 and 400 sequences.

- **Amazon**(Ni, 2018). This dataset includes time-stamped user product reviews behavior from January, 2008 to October, 2018. Each user has a sequence of produce review events with each event containing the timestamp and category of the reviewed product, with each category corresponding to an event type. We work on a subset of 5200 most active users with an average sequence length of 70 and then end up with $K = 16$ event types.

- **Retweet** (Ke Zhou & Song., 2013). This dataset contains time-stamped user retweet event sequences. The events are categorized into $K = 3$ types: retweets by "small," "medium" and "large" users. Small users have fewer than 120 followers, medium users have fewer than 1363, and the rest are large users. We work on a subset of 5200 active users with an average sequence length of 70.

- **Taxi** (Whong, 2014). This dataset tracks the time-stamped taxi pick-up and drop-off events across the five boroughs of the New York City; each (borough, pick-up or drop-off) combination defines an event type, so there are $K = 10$ event types in total. We work on a randomly sampled subset of 2000 drivers with an average sequence length of 39.

- **Taobao** (Xue et al., 2022). This dataset contains time-stamped user click behaviors on Taobao shopping pages from November 25 to December 03, 2017. Each user has a sequence of item click events with each event containing the timestamp and the category of the item. The categories of all items are first ranked by frequencies and the top 19 are kept while the rest are merged into one category, with each category corresponding to an event type. We work on a subset of 4800 most active users with an average sequence length of 150 and then end up with $K = 20$ event types.

- **StackOverflow** (Leskovec & Krevl, 2014). This dataset has two years of user awards on a question-answering website: each user received a sequence of badges and there are $K = 22$ different kinds of badges in total. We work on a subset of 2200 active users with an average sequence length of 65.

All preprocessed datasets are available at Google Drive.

**Evaluation Protocol.** We keep the model architectures as the original implementations in their papers. For a fair comparison, we use the same training procedure for all the models: we used the same optimizer (Adam (Kingma & Ba, 2015) with default parameters), biases initialized with zeros, no learning rate decay, the same maximum number of training epochs, and early stopping criterion (based on log-likelihood on the held-out dev set) for all models.

We mainly examine the models in two standard scenarios.

- Goodness-of-fit: we fit the models on the train set and measure the log-probability they assign to the held-out data.

- Next-event prediction: we use the minimum Bayes risk (MBR) principle to predict the next event time given only the preceding events, as well as its type given both its true time and the preceding events. We evaluate the time and type prediction by RMSE and error rate, respectively.

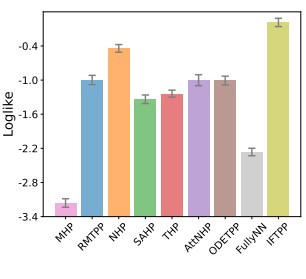 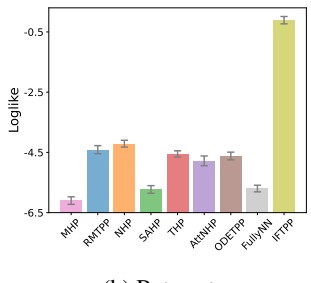 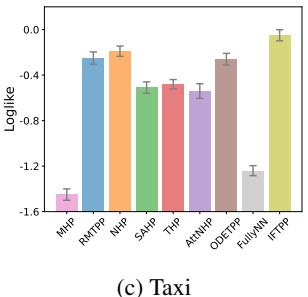

(a) Synthetic 1-D Hawkes         (b) Retweet         (c) Taxi

Figure 5: Performance of all the methods on the goodness-of-fit task on synthetic Hawkes, Retweet, and Taxi data. A higher score is better. All methods are implemented in PyTorch.

In addition, we propose a new evaluation task: the long-horizon prediction. Given the prefix of each held-out sequence $x_{[0,T]}$, we autoregressively predict the next events in a future horizon $\hat{x}_{(T,T']}$. It is evaluated by measuring the optimal transport distance (OTD), a type of edit distance for event sequences (Mei et al., 2019), between the prediction $\hat{x}_{(T,T']}$ and ground truth $x_{(T,T']}$. As pointed out by Xue et al. (2022), long-horizon prediction of event sequences is essential in various real-world domains, and this task provides new insight into the predictive performance of the models.

It is worth noting that FullyNN, faithfully implemented based on the author's version, does not support multi-type event sequences. Therefore it is excluded from the type prediction task.

### 5.2 Results and Analysis

**Main Results on Goodness-of-Fit and Next-Event Prediction.**

- Figure 5 reports the log-likelihood on three held-out datasets for all the methods. We find IFTPP outperforms all the competitors because it evaluates the log-likelihood in a close form while the others (RMTPP, NHP, THP, AttNHP, ODETPP) compute the intensity function via Monte Carlo integration, causing numerical approximation errors. FullyNN method, which also exactly computes the log-likelihood, has worse fitness than other neural competitors. As Shchur et al. (2020) points out, the PDF of FullyNN does not integrate to 1 due to a suboptimal choice of the network architecture, therefore causing a negative impact on the performance.

- Figure 6 reports the time and type prediction results on three real datasets. We find there is no single winner against all the other methods. Attention-based methods (SAHP, THP, AttNHP) generally perform better than or close to non-attention methods (RMTPP, NHP, ODETPP,FullyNN and IFTPP) on Amazon, Taobao, and Stackoverflow, while NHP is the winner on both Retweet and Taxi. We see that NHP is a comparably strong baseline with attention-based TPPs. This is not too surprising because similar results have been reported in previous studies (Yang et al., 2022).

- Not surprisingly, the performance of the classical model MHP is worse than the neural models across most of the evaluation tasks, consistent with the previous findings that neural TPPs have demonstrated to be more effective than classical counterparts at fitting data and making predictions.

Please see Appendix E.3 for the complete results (in numbers) on all the datasets. With a growing number of TPP methods proposed, we will continuously expand the catalog of models and datasets and actively update the benchmark in our Github repository.

**Analysis-I: Long Horizon Prediction.** We evaluate the long horizon prediction task on Retweet and Taxi datasets. On both datasets, we set the prediction horizon to be the one that approximately has 5 and 10 events, respectively. Shown in Figure 7 and Figure 8, we find that AttNHP and THP are two co-winners on Retweet and THP is a single winner on Taxi. Nonetheless, the margin of the winner over the competitors is small. The exact numbers shown in these two figures could be found in Table 5 in Appendix E.3. Due to the fact that these models are autoregressive and locally normalized,

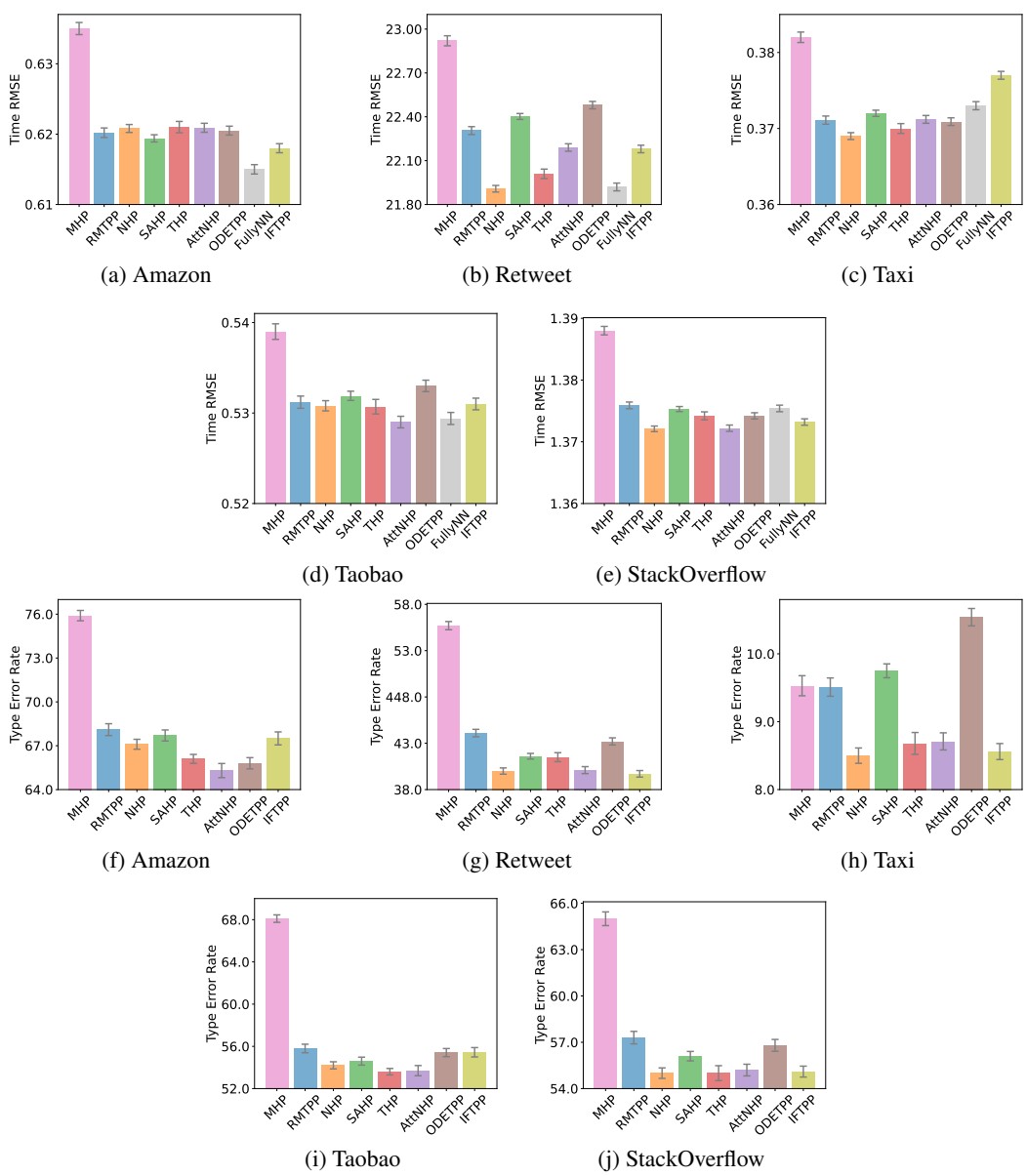

Figure 6: Performance of all the methods on next-event's time prediction (first row) and next-event's type prediction (second row) on five real datasets. Lower score is better. All methods are implemented in PyTorch. As clarified, FullyNN is not applicable for the type prediction tasks.

they are all exposed to cascading errors. To fix this issue, one could resort to globally normalized models (Xue et al., 2022), which is out of the scope of the paper.

**Analysis-II: Models with Different Frameworks: PyTorch vs. TensorFlow.** Researchers normally implement their experiments and models for specific ML frameworks. For example, recently proposed methods are mostly restricted to PyTorch and are not applicable to TensorFlow models. As explained in Section 4, to facilitate the use of TPPs, we implement two equivalent sets of methods in PyTorch and TensorFlow. Table 1 shows the relative difference between the results of Torch and TensorFlow implementations are all within $[-1.5\%, 1.5\%]$. To conclude, although the code could not be exactly the same, the two sets of models produce similar performance in terms of predictive ability.

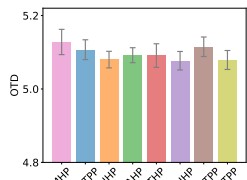 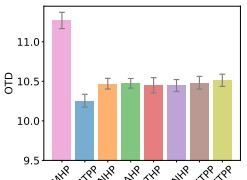 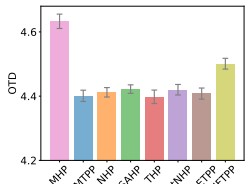 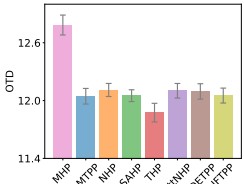

Figure 7: Long horizon prediction on Retweet data: left (avg prediction horizon 5 events) vs. right (avg prediction horizon 10 events).

Figure 8: Long horizon prediction on Taxi data: left (avg prediction horizon 5 events) vs. right (avg prediction horizon 10 events).

| MODEL | REL DIFF ON TIME RMSE (1ST ROW) AND TYPE ERROR RATE (2ND ROW) | | | | |
|---|---|---|---|---|---|
| | AMAZON | RETWEET | TAXI | TAOBAO | STACKOVERFLOW |
| RMTPP | −0.2% | +1.0% | +0.1% | +0.1% | +0.4% |
| | +0.5% | +1.3% | +0.6% | +0.2% | −0.7% |
| NHP | +0.7% | +0.5% | −0.2% | +0.1% | −0.1% |
| | +0.6% | +1.4% | +0.4% | −0.3% | −0.1% |
| SAHP | −0.8% | +0.7% | −0.8% | +0.4% | 0.3% |
| | +0.6% | +0.6% | −0.6% | +0.4% | 0.3% |
| THP | +0.6% | +0.6% | −0.2% | −0.5% | 0.6% |
| | +1.2% | +0.9% | −0.6% | +0.7% | 0.4% |
| ATTNHP | +0.4% | +0.4% | +0.3% | −0.1% | −0.2% |
| | +0.2% | −0.7% | −0.6% | +0.4% | +0.2% |
| ODETPP | −0.5% | +1.1% | +0.9% | +0.6% | 0.4% |
| | +0.8% | +1.3% | +1.1% | −0.5% | −0.5% |
| FULLYNN | +0.5% | −0.7% | −0.3% | −0.3% | +0.2% |
| | NA | NA | NA | NA | NA |
| IFTPP | −0.9% | +1.0% | +0.4% | +0.6% | +0.3% |
| | +0.4% | −0.7% | −0.3% | +0.2% | +0.2% |

Table 1: Relative difference between Torch and TensorFlow implementations of methods in Figure 6.

## 6  Future Research Opportunities

We summarize our thoughts on future research opportunities inspired by our benchmarking results.

Most importantly, the results seem to be signaling that we should think beyond architectural design. For the past decade, this area has been focusing on developing new architectures, but the performance of new models on the standard datasets seem to be saturating. Notably, all the best to-date models make poor predictions on time of future events. Moreover, on type prediction, attention-based models (Zuo et al., 2020; Zhang et al., 2020; Yang et al., 2022) only outperform other architectures by a small margin. Looking into the future, we advocate for a few new research directions that may bring significant contributions to the field.

The first is to build foundation models for event sequence modeling. The previous model-building work all learns data-specific weights, and does not test the transferring capabilities of the learned models. Inspired by the emergence of foundation models in other research areas, we think it will be beneficial to explore the possibility to build foundation models for event sequences. Conceptually, learning from a large corpus of diverse datasets—like how GPTs (Nakano et al., 2021) learn by reading open web text—has great potential to improve the model performance and generalization beyond what could be achieved in the current in-domain in-data learning paradigm. Our library can facilitate exploration in this direction since we unify the data formats and provide an easy-to-use interface that users can seamlessly plug and play any set of datasets. Challenges in this direction arise as different datasets tend to have disjoint sets of event types and different scales of time units.

The second is to go beyond event data itself and utilize external information sources to enhance event sequence modeling. Seeing the performance saturation of the models, we are inspired to think whether the performance has been bounded by the intrinsic signal-to-noise ratio of the event sequence data. Therefore, it seems natural and beneficial to explore the utilization of other information sources, which include but are not limited to: (i) sensor data such as satellite images and radiosondes signals; (ii) structured and unstructured knowledge bases (e.g., databases, Wikipedia, textbooks); (iii) large pretrained models such as ChatGPT (Brown et al., 2020) and GPT-4 (OpenAI, 2023), whose rich knowledge and strong reasoning capabilities may assist event sequence models in improving their prediction accuracies.

The third is to go beyond observational data and embed event sequence models into real-world interventions (Qu et al., 2023). With interventional feedback from the real world, an event sequence model would have the potential to learn real causal dynamics of the world, which may significantly improve prediction accuracy.

All the aforementioned directions open up research opportunities for technical innovations.

## 7 Related work

**Temporal Point Processes.** Over recent years, a large variety of TPP models have been proposed, many of which are built on recurrent neural networks (Du et al., 2016; Mei & Eisner, 2017; Xiao et al., 2017; Omi et al., 2019; Shchur et al., 2020; Mei et al., 2020; Boyd et al., 2020). Models of this kind enjoy continuous state spaces and flexible transition functions, thus achieving superior performance on many real-world datasets, compared to the classical Hawkes process (Hawkes, 1971). To properly capture the long-range dependency in the sequence, the attention and transformer techniques (Vaswani et al., 2017) have been adapted to TPPs (Zuo et al., 2020; Zhang et al., 2020; Yang et al., 2022; Wen et al., 2023) and makes further improvements on predictive performance. Despite significant progress made in academia, the existing studies usually perform model evaluations and comparisons in an ad-hoc manner, e.g., by using different experimental settings or different ML frameworks. Such conventions not only increase the difficulty in reproducing these methods but also may lead to inconsistent experimental results among them.

**Open Benchmarking on TPPs.** The significant attention attracted by TPPs in recent years naturally leads to a high demand for an open benchmark to fairly compare against baseline models. While many efforts have been made in the domains of recommender systems (Zhu et al., 2021), computer vision (Deng et al., 2009), and natural language processing (Wang et al., 2019), benchmarking in the field of TPPs is an under-explored topic. *Tick* (Bacry et al., 2017) and *pyhawkes* [5] are two well-known libraries that focus on statistical learning for classical TPPs, which are not suitable for the state-of-the-art neural models. *Poppy* (Xu, 2018) is a PyTorch-based toolbox for neural TPPs, but it has not been actively maintained since three years ago and has not implemented any recent state-of-the-art methods. To the best of our knowledge, EasyTPP is the first package that provides open benchmarking for the popular neural TPPs.

## 8 Conclusion

In this work, we presented EasyTPP, a versatile benchmarking platform for the standardized and transparent comparison of TPP methods on different integrated data sets. With a growing open-source community, EasyTPP has the potential to become the main library for benchmarking TPPs. The community seems to really appreciate this initiative: without any advertising, our library has collected around 90 stars on Github and has been downloaded around 700 times from PyPi since it was released 3 months ago. We hope that this work continuously contributes to further advances in the research.

---

[5] https://github.com/slinderman/pyhawkes.

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
