# Appendices

## A  Limitation and Societal Impacts

**Limitations.** Our framework mainly implements models with neural networks, which are known to be data-hungry. Although they worked well in our experiments, it might still suffer compared to non-neural models if starved of data.

**Societal Impacts.** By releasing the benchmarking code and data, we hope to facilitate the modeling of continuous-time sequential data in many domains. However, our method may be applied to unethical ends. For example, its abilities of better fitting data and making more accurate predictions could potentially be used for unwanted tracking of individual behavior, e.g. for surveillance.

## B  `EasyTPP`'s Software Interface Details

In this section, we describe the architecture of our open-source benchmarking software `EasyTPP` in more detail and provide examples of different use cases and their implementation.

### B.1  High Level Software Architecture

The purpose of building `EasyTPP` is to provide a simple and standardized framework to allow users to apply different state-of-the-art (SOTA) TPPs to arbitrary data sets. For researchers, `EasyTPP` provides an implementation interface to integrate new recourse methods in an easy-to-use way, which allows them to compare their method to already existing methods. For industrial practitioners, the availability of benchmarking code helps them easily assess the applicability of TPP models for their own problems.

A high level visualization of the `EasyTPP`'s software architecture is depicted in Figure 9. *Data Preprocess* component provides a common way to access the event data across the software and maintains information about the features. For the *Model* component, the library provides the possibility to use existing methods or extend the users' custom methods and implementations. A *wrapper* encapsulates the black-box models along with the trainer and sampler. The primary purpose of the wrapper is to provide a common interface to easily fit in the training and evaluation pipeline, independently of their framework (e.g., PyTorch, TensorFlow). See Appendix B.2 and Appendix B.3 for details. The running of the pipeline is parameterized by the configuration class - *RunnerConfig* (without hyper-parameter tuning) and *HPOConfig* (with hyper-parameter tuning).

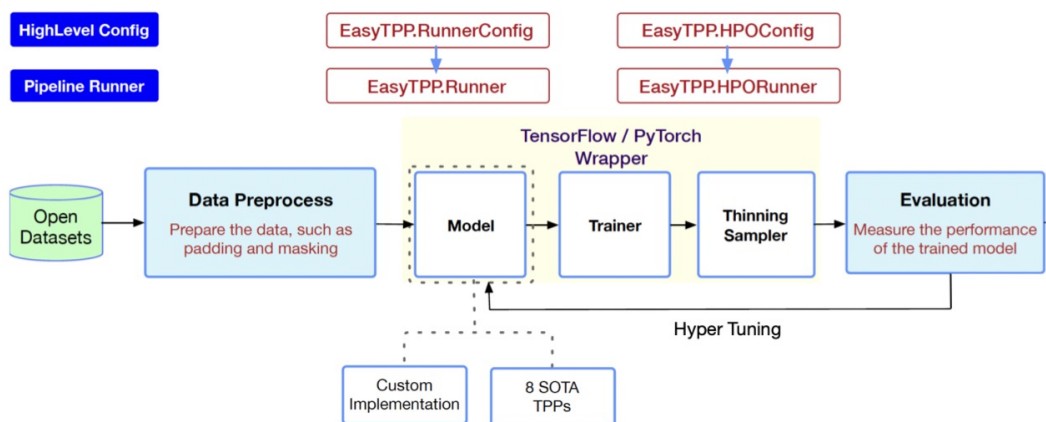

Figure 9: Architecture of the EasyTPP library. The dashed arrows show the different implementation possibilities, either to use pre-defined SOTA TPP models or provide a custom implementation. All dependencies between these objects are visualized by solid arrows with an additional description. The running of the pipeline is parameterized by the configuration classes - *RunnerConfig* (w/o hyper tuning) and *HPOConfig* (with hyper tuning).

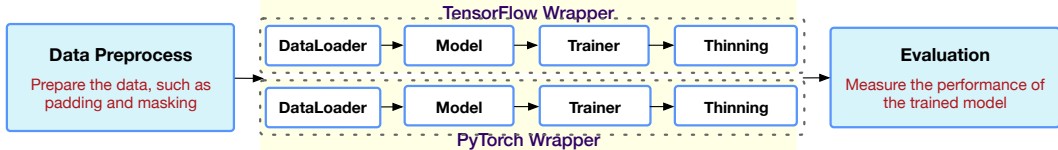

Figure 10: Illustration of TensorFlow and PyTorch Wrappers in the EasyTPP library.

## B.2 Why Does `EasyTPP` Support Both TensorFlow and PyTorch

TensorFlow and PyTorch are the two most popular Deep Learning (DL) frameworks today. PyTorch has a reputation for being a research-focused framework, and indeed, most of the authors have implemented TPPs in PyTorch, which are used as references by `EasyTPP`. On the other hand, TensorFlow has been widely used in real world applications. For example, Microsoft recommender,[6] NVIDIA Merlin[7] and Alibaba EasyRec[8] are well-known industrial user modeling systems with TensorFlow as the backend. In recent works, TPPs have been introduced to better capture the evolution of the user preference in continuous-time (Bao & Zhang, 2021; Fan et al., 2021; Bai et al., 2019). To support the use of TPPs by industrial practitioners, we implement an equivalent set of TPPs in TensorFlow. As a result, `EasyTPP` not only helps researchers analyze the strengths and bottlenecks of existing models, but also facilitates the deployment of TPPs in industrial applications.

## B.3 How Does `EasyTPP` Support Both PyTorch and TensorFlow

We implement two equivalent sets of data loaders, models, trainers, thinning samplers in TensorFlow and PyTorch, respectively, then use wrappers to encapsulate them so that they have the same API exposed in the whole training and evaluation pipeline. See Figure 10.

## B.4 `EasyTPP` for Researchers

The research groups can inherit from the *BaseModel* to implement their own method in `EasyTPP`. This opens up a way of standardized and consistent comparisons between different TPPs when exploring new models.

Specifically, if we want to customize a TPP in PyTorch, we need to initialize the model by inheriting the class *TorchBaseModel*:

```
from easy_tpp.model.torch_model.torch_basemodel import TorchBaseModel

# Custom Torch TPP implementations need to
# inherit from the TorchBaseModel interface
class NewModel(TorchBaseModel):
    def __init__(self, model_config):
        super(NewModel, self).__init__(model_config)

    # Forward along the sequence, output the states / intensities at the
    event times
    def forward(self, batch):
        ...
        return states

    # Compute the loglikelihood loss
    def loglike_loss(self, batch):
        ....
        return loglike
```

---

[6]https://github.com/microsoft/recommenders.
[7]https://developer.nvidia.com/nvidia-merlin.
[8]https://github.com/alibaba/EasyRec.

```
513  20
514  21      # Compute the intensities at given sampling times
515  22      # Used in the Thinning sampler
516  23      def compute_intensities_at_sample_times(self, batch, sample_times, **
517      kwargs):
518  24          ...
519  25          return intensities
```

Listing 1: Pseudo implementation of customizing a TPP model in PyTorch using EasyTPP.

Equivalent, if we want to customize a TPP in TensorFlow, we need to initialize the model by inheriting the class *TfBaseModel*:

```
522  1
523  2  from easy_tpp.model.torch_model.tf_basemodel import TfBaseModel
524  3
525  4  # Custom Torch TPP implementations need to
526  5  # inherit from the TorchBaseModel interface
527  6  class NewModel(TfBaseModel):
528  7      def __init__(self, model_config):
529  8          super(NewModel, self).__init__(model_config)
530  9
531  10      # Forward along the sequence, output the states / intensities at the
532      event times
533  11      def forward(self, batch):
534  12          ...
535  13          return states
536  14
537  15
538  16      # Compute the loglikelihood loss
539  17      def loglike_loss(self, batch):
540  18          ....
541  19          return loglike
542  20
543  21      # Compute the intensities at given sampling times
544  22      # Used in the Thinning sampler
545  23      def compute_intensities_at_sample_times(self, batch, sample_times, **
546      kwargs):
547  24          ...
548  25          return intensities
```

Listing 2: Pseudo implementation of customizing a TPP model in TensorFlow using EasyTPP.

## B.5  **EasyTPP** as a Modeling Library

A common usage of the package is to train and evaluate some standard TPPs. This can be done by loading black-box-models and data sets from our provided datasets, or by user-defined models and datasets via integration with the defined interfaces. Listing 3 shows an implementation example of a simple use-case, fitting a TPP model method to a preprocessed dataset from our library.

```
554  1  import argparse
555  2
556  3  from easy_tpp.config_factory import Config
557  4  from easy_tpp.runner import Runner
558  5
559  6
560  7  def main():
561  8      parser = argparse.ArgumentParser()
562  9
563  10      parser.add_argument('--config_dir',
564  11                          type=str,
565  12                          required=False,
566  13                          default='configs/experiment_config.yaml',
567  14                          help='Dir of configuration yaml to train and
568      evaluate the model.')
```

```
569  15
570  16    parser.add_argument('--experiment_id',
571  17                         type=str,
572  18                         required=False,
573  19                         default='IntensityFree_train',
574  20                         help='Experiment id in the config file.')
575  21
576  22    args = parser.parse_args()
577  23
578  24    # Build up the configuation for the runner
579  25    config = Config.build_from_yaml_file(args.config_dir, experiment_id=
580        args.experiment_id)
581  26
582  27    # Intialize the runner for the pipeline
583  28    model_runner = Runner.build_from_config(config)
584  29
585  30    # Start running
586  31    model_runner.run()
587  32
588  33
589  34 if __name__ == '__main__':
590  35     main()
```

Listing 3: Example implementation of running a TPP model using EasyTPP.

## C  Model Implementation Details

We have implemented the following TPPs

- **Recurrent marked temporal point process (RMTPP)** (Du et al., 2016)**.** We implemented both the Tensorflow and PyTorch version of RMTPP by our own.

- **Neural Hawkes process (NHP)** (Mei & Eisner, 2017) and **Attentive neural Hawkes process (AttNHP)** (Yang et al., 2022)**.** The Pytorch implementation mostly comes from the code from the public GitHub repository at `https://github.com/yangalan123/anhp-andtt` (Yang et al., 2022) with MIT License. We developed the Tensorflow version of NHP and ttNHP by our own.

- **Self-attentive Hawkes process (SAHP)** (Zhang et al., 2020) and **transformer Hawkes process (THP)** (Zuo et al., 2020)**.** We rewrote the PyTorch versions of SAHP and THP based on the public Github repository at `https://github.com/yangalan123/anhp-andtt` (Yang et al., 2022) with MIT License. We developed the Tensorflow versions of the two models by our own.

- **Intensity-free TPP (IFTPP)** (Shchur et al., 2020)**.** The Pytorch implementation mostly comes from the code from the public GitHub repository at `https://github.com/shchur/ifl-tpp` (Shchur et al., 2020) with MIT License. We implemented a Tensorflow version by our own.

- **Fully network based TPP (FullyNN)** (Omi et al., 2019)**.** We rewrote both the Tensorflow and PyTorch versions of the model faithfully based on the author's code at `https://github.com/omitakahiro/NeuralNetworkPointProcess`. Please not that the model only considers the number of the types to be one, i.e., the sequence's $K = 1$.

- **ODE-based TPP (ODETPP)** (Chen et al., 2021)**.** We implement a TPP model, in both Tensorflow and PyTorch, with a continuous-time state evolution governed by a neural ODE. It is basically the spatial-temporal point process (Chen et al., 2021) without the spatial component.

### C.1  Likelihood Computation Details

In this section, we discuss the implementation details of NLL computation in Equation (4).

The integral term in Equation (4) is computed using the Monte Carlo approximation given by Mei & Eisner (2017, Algorithm 1), which samples times $t$. This yields an unbiased stochastic gradient. For the number of Monte Carlo samples, we follow the practice of Mei & Eisner (2017): namely, at training time, we match the number of samples to the number of observed events at training time, a

619 reasonable and fast choice, but to estimate log-likelihood when tuning hyperparameters or reporting
620 final results, we take 10 times as many samples.

621 At each sampled time $t$, the Monte Carlo method still requires a summation over all events to obtain
622 $\lambda(t)$. This summation can be expensive when there are many event types. This is not a serious
623 problem for our EasyTPP implementation since it can leverage GPU parallelism.

624 ## C.2 Next Event Prediction

625 It is possible to sample event sequences exactly from any intensity-based model in EasyTPP, using
626 the **thinning algorithm** that is traditionally used for autoregressive point processes (Lewis & Shedler,
627 1979; Liniger, 2009). In general, to apply the thinning algorithm to sample the next event at time
628 $\geq t_0$, it is necessary to have an upper bound on $\{\lambda_e(t) : t \in [t_0, \infty)\}$ for each event type $t$. An
629 explicit construction for the NHP (or AttNHP) model was given by Mei & Eisner (2017, Appendix
630 B.3).

631 Section 3 includes a task-based evaluation where we try to predict the *time* and *type* of just the next
632 event. More precisely, for each event in each held-out sequence, we attempt to predict its time given
633 only the preceding events, as well as its type given both its true time and the preceding events.

634 We evaluate the time prediction with average $L_2$ loss (yielding a root-mean-squared error, or **RMSE**)
635 and evaluate the argument prediction with average 0-1 loss (yielding an **error rate**).

636 Following Mei & Eisner (2017), we use the minimum Bayes risk (MBR) principle to predict the time
637 and type with the lowest expected loss. For completeness, we repeat the general recipe in this section.

638 For the $i^{\text{th}}$ event, its time $t_i$ has density $p_i(t) = \lambda(t) \exp(-\int_{t_{i-1}}^{t} \lambda(t')dt')$. We choose $\int_{t_{i-1}}^{\infty} t p_i(t)dt$
639 as the time prediction because it has the lowest expected $L_2$ loss. The integral can be estimated using
640 i.i.d. samples of $t_i$ drawn from $p_i(t)$ by the thinning algorithm.

641 Given the next event time $t_i$, we choose the most probable type $\text{argmax}_e \lambda_e(t_i)$ as the type prediction
642 because it minimizes expected 0-1 loss.

643 ## C.3 Long Horizon Prediction

644 The TPP models are typically autoregressive: predicting each future event is conditioned on all the
645 previously predicted events. Following the approach in (Xue et al., 2022), we set up a prediction
646 horizon and use OTD to measure the divergence between the ground truth sequence and the predicted
647 sequence within the horizon. For more details about the setup and evaluation protocol, please see
648 Section 5 in Xue et al. (2022).

649 # D  Dataset Details

650 To comprehensively evaluate the models, we preprocessed one synthetic and five real-world datasets
651 from widely-cited works that contain diverse characteristics in terms of their application domains and
652 temporal statistics.

- **Synthetic.** This dataset contains synthetic event sequences from a univariate Hawkes process
  sampled using `Tick` (Bacry et al., 2017) whose conditional intensity function is defined by

$$\lambda(t) = \mu + \sum_{t_i < t} \alpha\beta \cdot \exp(-\beta(t - t_i))$$

653   with $\mu = 0.2, \alpha = 0.8, \beta = 1.0$. We randomly sampled disjoint train, dev, and test sets with 1200,
654   200 and 400 sequences.

- **Amazon** (Ni, 2018). This dataset includes time-stamped user product reviews behavior from
  656   January, 2008 to October, 2018. Each user has a sequence of produce review events with each event
  657   containing the timestamp and category of the reviewed product, with each category corresponding
  658   to an event type. We work on a subset of $5200$ most active users with an average sequence length
  659   of 70 and then end up with $K = 16$ event types.

| DATASET | $K$ | # OF EVENT TOKENS | | | SEQUENCE LENGTH | | |
|---|---|---|---|---|---|---|---|
| | | TRAIN | DEV | TEST | MIN | MEAN | MAX |
| RETWEET | 3 | 369000 | 62000 | 61000 | 10 | 41 | 97 |
| TAOBAO | 17 | 350000 | 53000 | 101000 | 3 | 51 | 94 |
| AMAZON | 16 | 288000 | 12000 | 30000 | 14 | 44 | 94 |
| TAXI | 10 | 51000 | 7000 | 14000 | 36 | 37 | 38 |
| STACKOVERFLOW | 22 | 90000 | 25000 | 26000 | 41 | 65 | 101 |
| HAWKES-1D | 1 | 55000 | 7000 | 15000 | 62 | 79 | 95 |

Table 2: Statistics of each dataset.

- **Retweet** (Ke Zhou & Song., 2013). This dataset contains time-stamped user retweet event sequences. The events are categorized into $K = 3$ types: retweets by "small," "medium" and "large" users. Small users have fewer than 120 followers, medium users have fewer than 1363, and the rest are large users. We work on a subset of 5200 most active users with an average sequence length of 70.

- **Taxi** (Whong, 2014). This dataset tracks the time-stamped taxi pick-up and drop-off events across the five boroughs of the New York City; each (borough, pick-up or drop-off) combination defines an event type, so there are $K = 10$ event types in total. We work on a randomly sampled subset of 2000 drivers and each driver has a sequence. We randomly sampled disjoint train, dev and test sets with 1400, 200 and 400 sequences.

- **Taobao** (Xue et al., 2022). This dataset contains time-stamped user click behaviors on Taobao shopping pages from November 25 to December 03, 2017. Each user has a sequence of item click events with each event containing the timestamp and the category of the item. The categories of all items are first ranked by frequencies and the top 19 are kept while the rest are merged into one category, with each category corresponding to an event type. We work on a subset of 4800 most active users with an average sequence length of 150 and then end up with $K = 20$ event types.

- **StackOverflow** (Leskovec & Krevl, 2014). This dataset has two years of user awards on a question-answering website: each user received a sequence of badges and there are $K = 22$ different kinds of badges in total. We randomly sampled disjoint train, dev and test sets with 1400, 400 and 400 sequences from the dataset.

Table 2 shows statistics about each dataset mentioned above.

# E  Experiment Details

## E.1  Setup

**Training Details.** For TPPs, the main hyperparameters to tune are the hidden dimension $D$ of the neural network and the number of layers $L$ of the attention structure (if applicable). In practice, the optimal $D$ for a model was usually $16, 32, 64$; the optimal $L$ was usually $1, 2, 3, 4$. To train the parameters for a given generator, we performed early stopping based on log-likelihood on the held-out dev set. The chosen parameters for the main experiments are given in Table 6.

**Computation Cost.** All the experiments were conducted on a server with 256G RAM, a 64 logical cores CPU (Intel(R) Xeon(R) Platinum 8163 CPU @ 2.50GHz) and one NVIDIA Tesla P100 GPU for acceleration. For training, the batch size is 256 by default. On all the dataset, the training of AttNHP takes most of the time (i.e., around 4 hours) while other models take less than 2 hours.

## E.2  Sanity Checks

For each model we reproduced in our library, we ran experiments to ensure that our implementation could match the results in the original paper. We used the same hyperparameters as in original papers; we reran each experiment 5 times and took the average.

In Table 3, we show the relative differences between the implementations on Retweet and Taxi datasets. As we can see, all the relative differences are within $(-5\%, 5\%)$, indicating that our implementation is close to the original.

| MODEL | METRICS (TIME RMSE / TYPE ERROR RATE) | |
|---|---|---|
| | RETWEET | TAXI |
| RMTPP | $-4.1\%/-3.5\%$ | $-2.9\%/-3.7\%$ |
| NHP | $+3.4\%/+3.1\%$ | $+2.6\%/+3.5\%$ |
| SAHP | $+1.3\%/+1.7\%$ | $+1.1\%/+1.2\%$ |
| THP | $+1.3\%/+1.8\%$ | $-1.6\%/+1.5\%$ |
| AttNHP | $+1.2\%/-1.0\%$ | $-1.2\%/-1.2\%$ |
| ODETPP | $-4.0\%/-3.9\%$ | $-4.3\%/-4.5\%$ |
| FullyNN | $-5.0\%$/N.A. | $-4.1\%$/N.A. |
| IFTPP | $+3.4\%/+3.1\%$ | $+3.9\%/+3.0\%$ |

Table 3: The relative difference between the results of EasyTPP and original implementations.

## E.3 More Results.

For better visual comparisons, we present the results in Figure 6, Figure 7 and Figure 8 also in the form of tables, see Table 4 and Table 5.

| MODEL | METRICS (TIME RMSE / TYPE ERROR RATE) | | | | |
|---|---|---|---|---|---|
| | AMAZON | RETWEET | TAXI | TAOBAO | STACKOVERFLOW |
| MHP | 0.635/75.9% | 22.92/55.7% | 0.382/9.53% | 0.539/68.1% | 1.388/65.0% |
| RMTPP | 0.620/68.1% | 22.31/44.1% | 0.371/9.51% | 0.531/55.8% | 1.376/57.3% |
| NHP | 0.621/67.1% | **21.90**/40.0% | **0.369**/**8.50%** | 0.531/54.2% | **1.372**/**55.0%** |
| SAHP | 0.619/67.7% | 22.40/41.6% | 0.372/9.75% | 0.532/54.6% | 1.375/56.1% |
| THP | 0.621/66.1% | 22.01/41.5% | 0.370/8.68% | 0.531/**53.6%** | 1.374/55.0% |
| AttNHP | 0.621/**65.3%** | 22.19/40.1% | 0.371/8.71% | **0.529**/53.7% | **1.372**/55.2% |
| ODETPP | 0.620/65.8% | 22.48/43.2% | 0.371/10.54% | 0.533/55.4% | 1.374/56.8% |
| FullyNN | **0.615**/NA | 21.92/NA | 0.373/NA | 0.529/NA | 1.375/NA |
| IFTPP | 0.618/67.5% | 22.18/**39.7%** | 0.377/8.56% | 0.531/55.4% | 1.373/55.1% |

Table 4: Performance in numbers of all methods mentioned in Figure 6.

| MODEL | OTD | | | |
|---|---|---|---|---|
| | RETWEET AVG 5 EVENTS | RETWEET AVG 10 EVENTS | TAXI AVG 5 EVENTS | TAXI AVG 10 EVENTS |
| MHP | 5.128 | 11.270 | 4.633 | 12.784 |
| RMTPP | 5.107 | 10.255 | 4.401 | 12.045 |
| NHP | 5.080 | 10.470 | 4.412 | 12.110 |
| SAHP | 5.092 | 10.475 | 4.422 | 12.051 |
| THP | 5.091 | **10.450** | **4.398** | **11.875** |
| AttNHP | **5.077** | 10.447 | 4.420 | 12.102 |
| ODETPP | 5.115 | 10.483 | 4.408 | 12.095 |
| FullyNN | NA | NA | NA | NA |
| IFTPP | 5.079 | 10.513 | 4.501 | 12.052 |

Table 5: Long horizon prediction on Retweet and Taxi data.

| MODEL | DESCRIPTION | VALUE USED |
|---|---|---|
| | $hidden\_size$ | 32 |
| | $time\_emb\_size$ | 16 |
| RMTPP | $num\_layers$ | 2 |
| | $hidden\_size$ | 64 |
| | $time\_emb\_size$ | 16 |
| NHP | $num\_layers$ | 2 |
| | $hidden\_size$ | 32 |
| | $time\_emb\_size$ | 16 |
| SAHP | $num\_layers$ | 2 |
| | $num\_heads$ | 2 |
| | $hidden\_size$ | 64 |
| | $time\_emb\_size$ | 16 |
| THP | $num\_layers$ | 2 |
| | $num\_heads$ | 2 |
| | $hidden\_size$ | 32 |
| | $time\_emb\_size$ | 16 |
| ATTNHP | $num\_layers$ | 1 |
| | $num\_heads$ | 2 |
| | $hidden\_size$ | 32 |
| ODETPP | $time\_emb\_size$ | 16 |
| | $num\_layers$ | 2 |
| | $hidden\_size$ | 32 |
| FULLYNN | $time\_emb\_size$ | 16 |
| | $num\_layers$ | 2 |
| | $hidden\_size$ | 32 |
| INTENSITYFREE | $time\_emb\_size$ | 16 |
| | $num\_layers$ | 2 |

Table 6: Descriptions and values of hyperparameters used for models.

## F  Additional Note

### F.1  Citation Count in ArXiv

We search the TPP-related articles in ArXiv `https://arxiv.org/` using their own search engine in three folds:

- Temporal point process: we search through the abstract of articles which contains the term 'temporal point process'.
- Hawkes process: we search through the abstract of articles with the term 'hawkes process' but without the term 'temporal point process'.
- Temporal event sequence: we search through the abstract of articles which include the term 'temporal event sequence' but exclude the term 'hawkes process' and 'temporal point process'.

We group the articles found out by the search engine by years and report it in Figure 2.