# OpenReview forum: "EasyTPP: Towards Open Benchmarking the Temporal Point Processes"
_NeurIPS.cc/2023/Track/Datasets_and_Benchmarks — Submitted to NeurIPS 2023 Datasets and Benchmarks_

### Official Review · Reviewer_mYWh · 2023-07-11
**The code implementation is great. But the analysis could be improved.**

**Rating:** 5
**Confidence:** 4
**Correctness:** Yes
**Clarity:** Yes

**Strengths:**

- The benchmarking of TPP could facilitate reproducibility and accelerate domain research.
- The code implementation is consistent and comprehensive. It supports both tensorflow and torch.


**Additional Feedback:**

NA

**Documentation:**

Yes

**Limitations:**

Yes

**Opportunities For Improvement:**

1. As a user, I would like to know whether this benchmark's reimplemented methods are good reproduction of the original paper's code, i.e., sanity check. Please explain the reproducibility results under controlled settings of the original works.
2. By benchmarking, what's **new** do we know about TPP? Any new discovery or conflictive conclusions w.r.t literature?
3. More suggestions to be given to people entering the domain and people who are already researchers in TPP (if question 1 is solved, this could make more sense)
4. (minor) The comparison of TF and Torch is not quite clear in Figure 8. I would suggest revising the figure for visual comparison

**Relation To Prior Work:**

Yes

**Summary And Contributions:**

The paper presents EasyTPP, a comprehensive benchmark for evaluating temporal point processes (TPPs) in continuous-time event sequences. TPPs are widely used to model data in various domains, but the lack of standardization in evaluating TPP models hinders progress in the field. EasyTPP addresses this issue by providing an open-source benchmark that includes eight highly cited neural TPP models, commonly used evaluation metrics, and datasets.

The contributions of EasyTPP are threefold. Firstly, it offers a comprehensive implementation of neural TPP models with integrated evaluation metrics and datasets. This allows researchers and practitioners to compare different methods on a standardized platform. Secondly, EasyTPP establishes a benchmarking pipeline that ensures transparency and thorough comparison of methods across different datasets. This facilitates reproducibility and reliable evaluation of TPP models. Thirdly, EasyTPP supports multiple machine learning libraries such as PyTorch and TensorFlow, as well as custom implementations, making it a versatile and extensible framework.

The paper emphasizes the importance of a standardized benchmarking platform for neural TPPs to promote reproducible research and accelerate progress in the field. EasyTPP aims to provide a common ground for comparing different methods and selecting the most effective models for real-world applications. The benchmark is open-sourced and actively maintained, welcoming contributions from researchers and practitioners.

---

> ### Author Response · Authors · 2023-08-18
>
>
> > As a user, I would like to know whether this benchmark's reimplemented methods are good reproduction of the original paper's code, i.e., sanity check. Please explain the reproducibility results under controlled settings of the original works.
>
> Definitely! We took great care to make sure that our reimplementation is close to the original implementation. For the models with official open-sourced code, we did our best to match all the technical details. The source of the code is explained in Appendix B.
>
> For this rebuttal, we present an empirical comparison between our reimplementations vs. original code on two datasets. Please see [Sanity Check] for details.
>
> > By benchmarking, what's new do we know about TPP? Any new discovery or conflictive conclusions w.r.t literature?
>
> Thank you for raising this great question! Do our arguments in [New Analysis] answer your question?
> Please let us know if you think more analysis is needed.
>
> > More suggestions to be given to people entering the domain and people who are already researchers in TPP (if question 1 is solved, this could make more sense)
>
> We will restructure our presentation to facilitate reading and understanding of readers, making it more accessible to novices and more juicy for experienced researchers.
>
> Please see [Presentation Improvements] for our detailed edits.
>
>
> > (minor) The comparison of TF and Torch is not quite clear in Figure 8. I would suggest revising the figure for visual comparison
>
> For better visual comparison, we augment a table to show the relative differences (i.e., (torch result - tf result)/torch result, in %) for models illustrated in Figure 5 and Figure 8.
>
> Please see Table-4 on page-10 of the updated paper.

---

> > ### Author Response · Authors · 2023-08-29
> > **Follow up about paper revision.**
> >
> > > > (minor) The comparison of TF and Torch is not quite clear in Figure 8. I would suggest revising the figure for visual comparison
> >
> > > For better visual comparison, we augment a table to show the relative differences (i.e., (torch result - tf result)/torch result, in %) for models illustrated in Figure 5 and Figure 8.
> >
> > As we have updated paper after 08-18, the augmented table now corresponds to Table 1 at Page 9 of the most recent version.
> >
> > Please let us know if you have any remaining concerns.

---

> ### Author Response · Authors · 2023-08-28
> **Have we resolved your concerns?**
>
> Thank you again for your constructive feedback.
>
> In the rebuttal, we have tried to address all your concerns:
> - [reproducibility] We presented sanity check results.
> - [what's new] We added new analysis.
> - [audience] We reorganized the paper to improve reading experience of both beginners and experienced researchers.
> - [TF vs. Torch] We revised the presentation of these results.
>
> As our author-reviewer discussion is coming to its end, we are looking forward to knowing whether our responses have resolved all your concerns so far, and whether you would like to adjust your evaluation.
>
> We would really appreciate it if our next round of communication could leave enough time for us to resolve any of your remaining or new questions.

---

### Official Review · Reviewer_3bs8 · 2023-07-19
**Review of Submission 302**

**Rating:** 7
**Confidence:** 4

**Strengths:**

1. The authors provide a detailed analysis of the datasets, including their characteristics and the challenges they pose for temporal point process modeling. The authors also discuss the performance of different models on the datasets, providing insights into their respective strengths and weaknesses.
2. The datasets used in this work are highly relevant to the field of temporal point process modeling and provide a valuable resource for researchers in this area.
3. The authors conduct a comprehensive benchmark analysis of different temporal point process models on the datasets, providing a standardized and transparent comparison of their performance.
4. This work makes significant contributions to the field of temporal point process modeling by providing a versatile and comprehensive benchmark platform. The quality of the research is high, and the benchmark results provide valuable insights for researchers and practitioners alike. The open-sourcing of the datasets and benchmark code also has the potential to have a significant impact on the field.


**Additional Feedback:**

None

**Clarity:**

The authors clearly introduce the problem, provide relevant background information, and describe their approach in a logical and organized manner. The methods and results are presented clearly, with appropriate use of tables and figures to convey information. The paper also includes a clear and concise conclusion summarizing the main findings and contributions of the work. Overall, the writing is clear and easy to follow, with appropriate use of technical terminology.

**Correctness:**

The evaluation methods and experiment design appropriate and performed correctly.

**Documentation:**

The paper provides some information on data collection and organization, but more details on dataset size, diversity, and processing methods would enhance its rigor. The publicly available and open-sourced datasets used in this work could benefit from additional documentation on intended uses, hosting, licensing, and maintenance plans.

This work's high-quality research and versatile benchmark platform contribute significantly to the field of temporal point process modeling, with the potential to accelerate progress in the area through open-sourcing of the datasets and benchmark code.


**Limitations:**

See the above contents for improvement.

**Opportunities For Improvement:**

The dataset summarized in this article lacks natural event datasets, such as earthquake datasets and other non-anthropogenic datasets.

**Relation To Prior Work:**

The authors do provide a thorough literature review and discuss how their work builds upon previous research. They also highlight the limitations of previous approaches and the need for a standardized benchmark platform, which suggests that their work is addressing a gap in the existing literature.

**Summary And Contributions:**

The paper introduces EasyTPP, a central benchmark for evaluating temporal point process models (TPPs) in various real-world domains. EasyTPP provides a comprehensive implementation of eight neural TPPs with commonly used evaluation metrics and datasets, a standardized benchmark pipeline, and a universal framework supporting multiple ML libraries. The unique contributions of EasyTPP are its transparent and thorough comparisons of different methods on different datasets, open-source accessibility, and potential to promote reproducible research and accelerate progress in the field.

---

> ### Author Response · Authors · 2023-08-18
>
> > The dataset summarized in this article lacks natural event datasets, such as earthquake datasets and other non-anthropogenic datasets.
>
> We will surely add such datasets. As explained in [Commitment], they are already in our Todo List (on Github page of our library).
>
> Particularly, we will add the following datasets:
> - Earthquake events. It is already used in a few papers such as THP (Zuo et al, ICML 2019). The source data is available in governmental earthquake data centers (China and US).
> - Volcano eruption events. The events correspond to the time-stamped volcano eruptions along with descriptive properties of the volcano as well as data related to economic and human impact of the eruption.  The data was sourced from the NOAA Significant Volcanic Eruption Database and it is available on the Kaggle website.

---

> > ### Comment · Reviewer_3bs8 · 2023-08-26
> >
> > Thanks for your response to my concern. I will keep my score as 7 and tend to accept this paper.

---

> > > ### Author Response · Authors · 2023-08-26
> > > **Thank you!**
> > >
> > > Thank you for your continued support!

---

### Official Review · Reviewer_JyMg · 2023-07-21
**Good paper, accept**

**Rating:** 7
**Confidence:** 2

**Strengths:**

**[Significance of the contribution]**
Standardization of libraries, great documentation.

**[Relevance to the broader research community]**
High.

**[Quality of the research]**
In general good.

**[Ethical and social implications]**
NA.


**Additional Feedback:**

NA.

**Clarity:**

**[Structure]**
Good.

**[Text]**
- Section 3: I can use some example (maybe in appendix) to understand the basics of TPPs.
- l.196, please clarify how the formula is composed of and what each part means, as it is not of general knowledge. I think the parts after $exp$ should be in superscript.
- General remark about footnotes. They should always come after punctuation and without spacing inbetween, e.g, line 173, line 220.
- Write the coferences of elements (sections, tables) in the format of Section X: line 271, Section 5.


**[Table]**
NA.

**[Figure]**
- In Figure 3, it is not clear how the loop from "Hyper Tuning" to "Model" is built in the stacks?
- Figure 5: Please add markers to the bars, as the colors are not black-white-print friendly. As it is hard to compare the bars in individual plots, I would suggest the authors to either put them in tables and highlight the numbers, or put the plots in one big plot, with all the bars to be compared next to each other (e.g., bars grouped by model and marked by dataset).
- Figures 6-7: Double check the title casing. Make sure it is consistent across titles.
- Rethink the presentation of Figures 6-7 as commented above.

**Correctness:**

**[Benchmark evaluation methods and experiment design appropriate and performed correctly?]**
Seemingly so.

**Documentation:**

**[Datasets with sufficient detail on data collection and organization, availability and maintenance, and ethical and responsible use? ]**
Good.

**[Dataset submissions include documentation and intended uses; a URL for reviewer access to the dataset; and a hosting, licensing, and maintenance plan? ]**
Good.

**[Benchmarks with sufficient detail to support reproducibility?]**
The github repo is well done.

**Ethics:**

**[Concerns are discussed?]**
NA.

**[Datasets' consideration of consent and privacy, responsible use, and legal compliance?]**
NA.

**Limitations:**

**[Addressed the limitations and potential negative societal impact of their work]**
NA.

**[Constructive suggestions for improvement]**
See above in "Opportunities For Improvement".



**Opportunities For Improvement:**

**[Significance of the contribution]**
- As I am not familiar with the domain, I quickly searched the open benchmarks on TPPs. I managed to find "Meta Temporal Point Processes", https://openreview.net/forum?id=QZfdDpTX1uM, which seems to be a paper the authors should have reviewed.


**Relation To Prior Work:**

**[Review of prior work]**
Good.

**[Is it clearly discussed how this work differs from previous contributions?]**
See "Opportunities For Improvement".


**Summary And Contributions:**

**[Summary]**
The paper has proposed the first TPP benchmark (seemingly claimed by the authors) to evaluate the neural network based TPPs. The authors have implemented eight highly cited nerual TPPs with common evaluations, and built standardized Tensorflow and PyTorch frameworks to support both academia and industries.

**[Contributions]**
- (C1): The authors cover eight representative neural TPPs over 5 standard datasets and 1 synthetic dataset.
- (C2): The authors have made great efforts to create frameworks in both Tensorflow and PyTorch to support further research in this direction.

---

> ### Author Response · Authors · 2023-08-18
>
> Thank you for your constructive feedback and being supportive. We will respond to your concerns below.
>
> > As I am not familiar with the domain, I quickly searched the open benchmarks on TPPs. I managed to find "Meta Temporal Point Processes", https://openreview.net/forum?id=QZfdDpTX1uM, which seems to be a paper the authors should have reviewed.
>
> Thanks for the reference! This new work introduces meta learning into training TPPs. As explained in [Commitment], we have added it to our Todo List and will implement it in our EasyTPP library. Its topic---meta learning---is also related to the future research directions discussed in [New Analysis], so we will cite and discuss it in the Future Research Opportunities section of the final version.
>
> > use some example (maybe in appendix) to understand the basics of TPPs.
>
> Definitely. As proposed in [Presentation Improvements], we will add visual illustration of data and basic technical concepts, which hopefully facilitates reading and understanding.
>
>
> > l.196, please clarify how the formula is composed of and what each part means, as it is not of general knowledge. I think the parts after exp should be in superscript.
>
> Thanks for pointing it out. The parts after exp indeed should be in superscript. It is fixed in the newly uploaded draft.
>
> L.196 describes the intensity function for an univariate Hawkes process
> - $\mu$ is the base intensity. When an event occurs, the intensity is elevated to some degree, but then will decay toward the base intensity $\mu$.
> - $\alpha \beta \exp \left(-\beta (t-t_i)\right)$ refers to the exponential kernel that describes how the intensity evolves when an event happens. Specifically, $\alpha \beta$  is the degree to which an event initially excites the intensity and $\beta$ is the decay rate (in exponential form) of that excitation.
> The summation operation means the effects of events are accumulated by time on the intensity.
>
> The more detailed explanation can be found either in the documentation of Tick library or section 3.1 of Neural Hawkes Process (Mei et al, NeurIPS, 2017).
>
>
>
> > General remark about footnotes. They should always come after punctuation and without spacing in between, e.g, line 173, line 220.
> >  Write the conferences of elements (sections, tables) in the format of Section X: line 271, Section 5.
>
> Thanks. Our updated version has this fixed.
>
> > In Figure 3, it is not clear how the loop from "Hyper Tuning" to "Model" is built in the stacks?
>
> Hyper Tuning works as a wrapper over the ‘Model’: it takes the model, hyperparameter space and search strategies to automate the efficient hyperparameter search based on the performance of evaluation tasks. We agree that a better illustration is we make the loop arrow from “Evaluation” point to “Model” directly and annotate it as the ‘Hyper Tuning’ process.
>
> > Figure 5: Please add markers to the bars, as the colors are not black-white-print friendly. As it is hard to compare the bars in individual plots, I would suggest the authors to either put them in tables and highlight the numbers, or put the plots in one big plot, with all the bars to be compared next to each other (e.g., bars grouped by model and marked by dataset).
>
> We found adding markers does not improve visibility. Per your advice, we add Table 2 containing all numbers, on page-10 of updated version.
>
>
>
>
> > Figures 6-7: Double check the title casing. Make sure it is consistent across titles.
>
> Fixed in the updated version.
>
> > Rethink the presentation of Figures 6-7 as commented above.
>
> Similarly, we augment a table to it.
> Please see Table-3 on page-10 of the updated paper.

---

> > ### Comment · Reviewer_JyMg · 2023-08-28
> >
> > Thank you for updating the paper with respect to the comments and marking the changes green.
> >
> >
> > Some updates I am not able to find, e.g.,
> > > "We found adding markers does not improve visibility. Per your advice, we add Table 2 containing all numbers, on page-10 of updated version."
> >
> > I did not manage to find Table 2.
> >
> > > "Similarly, we augment a table to it. Please see Table-3 on page-10 of the updated paper."
> >
> > I did not manage to find Table 3.

---

> > > ### Author Response · Authors · 2023-08-28
> > >
> > > Thank you very much for your feedback.
> > >
> > > After we replied on 08-18, we have updated the paper twice so that the table index has changed. Apologize for not timely informing you these changes.
> > >
> > > > I did not manage to find Table 2.
> > >
> > > This now corresponds to Table 4 in Appendix E.3 (Supplementary Material).
> > >
> > > > I did not manage to find Table 3.
> > >
> > > This now corresponds to Table 5 in Appendix E.3.
> > >
> > > Please let us know if our responses address your remaining concerns.

---

### Official Review · Reviewer_Q2cp · 2023-07-23

**Rating:** 6
**Confidence:** 4
**Clarity:** The paper is overall clear and easy t…

**Strengths:**

S1. This paper presents a unified benchmarking pipeline for evaluating neural temporal point process (TPP) models.
- Previous works on TPPs often make different choices in their evaluation settings, such as the embedding size and the evaluation metrics. These differences make it difficult to compare these methods in terms of their published results. This paper tackles this issue by developing a comprehensive benchmark environment for neural TPP models, where different models can be compared under the same settings.

S2. Extensive experimental results are provided in the paper.
- Eight recent TPP models are evaluated over six real-world and synthetic datasets, using three evaluation tasks.

S3. The writing is clear and easy to follow.

**Additional Feedback:**

Typos
- “with he fully”
- “protocal” (in appendix)

**Correctness:**

Yes. The experiments used several recent neural TPP methods, and the evaluation protocol follows a common practice used by previous studies.

**Documentation:**

This paper provides sufficient details for reproducing the benchmark results.

**Ethics:**

I do not think there will be ethical concerns with this work.

**Limitations:**

The paper discusses the limitations and potential negative societal impact of the proposed benchmark.

**Opportunities For Improvement:**

W1. The novelty and contributions of this work are somewhat limited.
- No new benchmark datasets or evaluation tasks are developed in this work. Both the datasets and the evaluation tasks used in this work to benchmark TPP methods have been used by previous studies.

W2. The TPP models included and evaluated in the proposed benchmark are all neural TPPs.
- There exist many classical TPP models that predate neural TPP methods, and these classical TPPs have advantages that neural TPPs do not have. The ability to compare against representative classical TPP models would also be a useful feature as a general benchmark for TPP models.

W3. Some applicable TPP models are excluded from evaluation.
- The IFTPP model was excluded from the prediction task as it does not explicitly calculate the intensity. However, IFTPP can be applied to the prediction task. Actually, the IFTPP paper performed an evaluation using a similar type of temporal prediction task.
- FullyNN seems applicable to the event time prediction, even if it cannot be used for event type prediction in its original form.

W4. The analysis of experimental results in Sec 6.2 can be improved.
- In addition to summarizing model performances, the analysis section can be further improved with, e.g., the discussion on some major limitations of existing TPP modeling paradigm, as well as opportunities for future research and improvement.

**Relation To Prior Work:**

Section 2 discusses the relation to prior work: compared to the few previous libraries that also provide implementations of TPP models, this work provides more recent neural TPP models.

**Summary And Contributions:**

This paper presents a benchmark for evaluating temporal point process (TPP) models. It provides a unified benchmarking pipeline for TPP models, with standard evaluation protocols, and the implementations of popular neural TPP models. Extensive experimental results are also given, where eight neural TPP models are evaluated on six datasets.

---

> ### Author Response · Authors · 2023-08-18
>
> Thank you for constructive feedback! We have added new results and new discussion for you; please see our to-all messages. Now we address each of your remaining concerns.
>
> > The novelty and contributions of this work are somewhat limited. No new…
>
> We are sorry that the submitted version didn't give a clear introduction to our contributions.
> Please see [Contribution and Significance] for our clarification. We will revise the final version accordingly.
>
>
> > TPP models included and evaluated in the proposed benchmark are all neural TPPs.
> > Some applicable TPP models are excluded from evaluation.
>
> We did so because the current state-of-the-art are all neural models. But we agree that we should expand the kinds of models evaluated in our benchmark.
>
> Please see [New Results] for the comparison of neural TPPs vs. other models (classical TPPs, intensity-free TPPs, etc).
>
> > The analysis of experimental results in Sec 6.2 can be improved.
> > the discussion on some major limitations of existing TPP modeling paradigm, as well as opportunities for future research and improvement
>
> Thanks for your suggestions! We thought about this but left such discussion out due to page limit. We will definitely add them back to the final version.
>
> Please see [New Analysis] for our thoughts on limitations of current TPP paradigm as well as possible future research directions.
>
> > Typos
>
> Thanks! Our updated version has fixed these issues.

---

> > ### Comment · Reviewer_Q2cp · 2023-08-25
> >
> > Thank you for the response. Authors made several improvements, including addtional experiments, new discussions, and paper revisions, which addressed several of my concerns. In light of this, I have increased my score.
> >
> > ---
> > Figure 1 (in the revised version) needs corrections.
> > - "future ensitie of the two types of is": typo
> > - hidden state (blue): cannot find this
> >
> > Also, code comments in Appendix are invisible.

---

> > > ### Author Response · Authors · 2023-08-26
> > > **Thank you!**
> > >
> > > Thank you very much! We are excited to hear that our discussion has led to your positive evaluation of the work.
> > >
> > > We have uploaded a new version, which includes the new corrections.

---

### Author Response · Authors · 2023-08-18
**New Results**

We ran new experiments to address concerns of reviewers. We first post all the new results in the tables below (with comparison to NHP, a strong neural model) and then discuss each of them in the following subsections (with titles [New Results - X]). In these tables, we add NHP as a competing model to benchmark the performances of the new models.

Table-7: log-likelihood on each dataset. (Table ID continues from the above reply)
|                         |   Retweet    | Taobao    |              Amazon |            Taxi |   Stackoverflow |
|-----------------|-----------------|--------------|----------------|---------------|---------------------|
| classical-MHP  |        -6.10     |       -0.37     |    -3.02            |   -1.45       |     -3.31      |
|   IFTPP             |         -0.11     |        -0.14    |      -1.95          |   -0.05      |     -1.07     |
|   FullyNN           |         -5.62     |        -0.49    |      -2.70          |   -1.24      |     -2.98     |
|   NHP               |         -4.21     |        -0.29    |      -2.64          |   -0.19      |     -2.90     |


Table-8: error rate of event type prediction
|            |   Retweet    | Taobao    |    Amazon |  Taxi |   Stackoverflow |
|------------------|-------------|--------------|-----------|--------|---------------|
| classical-MHP  |   55.7%  |   68.1%  |    75.9%    |   9.52%  |   65.0%    |
| IFTPP               |    39.5%  |   54.4%  |    67.5%     |   8.56%  |     55.1%   |
|   FullyNN          |    NA       |    NA       |      NA          |   NA       |    NA         |
|   NHP               |   40.0%  |  54.3%    |      67.1%    |   8.49%  |     54.9%  |



Table-9: RMSE of event time prediction
|                          |   Retweet    | Taobao    |              Amazon |                Taxi |   Stackoverflow |
|-----------------|----------|--------------|------------------------|-----------------|-------------|
| classical-MHP  |        22.922     |        0.590    |      0.715          |   0.439       |     1.448     |
| IFTPP              |         22.187     |        0.525    |      0.618          |   0.377       |     1.373     |
| FullyNN            |      21.925     |        0.529      |      0.615          |   0.368         |     1.375     |
|   NHP               |        21.892     |        0.531    |      0.621          |   0.370       |     1.374     |

Table-10: OTD for long-horizon prediction (avg over a horizon of 5 future events)
|                          |   Retweet    | Taxi         |
|------------------|---------------|-------------|
| classical-MHP  |        5.128  |        4.933    |
| IFTPP              |        5.079  |        4.501    |
| FullyNN           |           NA    |           NA    |
| NHP                 |        5.071  |        4.408    |

Note: original version of FullyNN by Omi et al. NeurIPS 2019 is not applicable for type prediction or long-horizon prediction.

[New Results - classical TPP]

R.Q2cp requests for results of "classical TPPs". We agree that "classical TPPs" are an important family of tools, but they have been covered by other libraries like tick and PoPPy. This benchmark focuses on neural TPPs mainly because neural models have demonstrated to be more effective than classical TPPs (at fitting data and making predictions). In addition, the research on neural TPPs has been making rapid progress, and we feel urgent to create a central benchmark for the line of work.

To further support our points, we fit a multivariate Hawkes process (a widely used and strong classical TPP) using tick, and compare it with neural TPPs; please see the tables above. Not surprisingly, their performance is worse than the neural model NHP across all the evaluation methods.

[New Results - intensity-free TPP]

Intensity-free TPP (IFTPP) is in the first release of our EasyTPP library, but we didn't evaluate it on predictive tasks since it is technically not a "point process". However, we did have plans to incorporate them in the future (as we explained in [Commitment]). Taking this opportunity, we ran new experiments for its prediction performance; please see tables above for results of the specific IFTPP by Shchur et al, ICLR 2020.

As we can see, IFTPP performs competitively to NHP: its error rate/RMSE is generally on par with that of NHP.

To enable MBR predictions of this kind of event sequence models, we implemented new modules in our EasyTPP library, further extending its potential impact.

[New Results - FullyNN]

Similar to the case of IFTPP, we implemented FullyNN in our library but didn't evaluate it on prediction tasks. Per reviewers' requests, we ran new experiments and presented new results of FullyNN in the above tables. As we can see, FullyNN has an overall narrow win over NHP on time prediction, but it is not capable of doing type prediction.

To enable time and type prediction of this model, we added new modules to our library, extending its potential impact.

---

### Author Response · Authors · 2023-08-18
**Sanity Check**

For each model we reproduced in our library, we ran experiments to ensure that our implementation could match the results in the original paper. In Table-5 and 6, we show the relative differences (i.e., (our result - original result)/original result, in %) between the implementations on Retweet and Taxi datasets. As we can see, all the relative differences are within (-5%, 5%), indicating that our implementation is close to the original.

Note: we used the same hyperparameters as in original papers; we reran each experiment 5 times and took the average.

Table-5: our implementation vs. original on Retweet ((Table ID continues from the paper)
|                 |  Err rate   | time RMSE     |
|------------|--------------|------------------------|
| NHP        |     +3.1%    |      +3.4%          |
| RMTPP   |    -3.5%     |      -4.1%          |
| THP         |   +1.8%     |      +1.3%          |
| SAHP      |    +1.7%     |      +1.3%         |
| ANHP      |    -1.0%     |       +1.2%         |
| ODETPP |   -3.9%     |         -4.0%        |
| IFTPP      |   +3.1%     |        +3.4%         |
| FullyNN   |    NA       |        -5.0%          |

Table-6: our implementation vs. original on Taxi
|                 |  Err rate   | time RMSE     |
|-------------|-----------|--------------------------|
| NHP        |     +3.5%    |      +2.6%          |
| RMTPP   |    -3.7%     |      -2.9%          |
| THP         |   +1.5%     |      -1.6%          |
| SAHP      |    +1.2%     |       +1.1%         |
| ANHP      |    -1.2%     |       -1.2%         |
| ODETPP |   -4.5%     |        - 4.3%        |
| IFTPP      |   +3.0%     |        +3.9%         |
| FullyNN   |    NA       |        -4.1%          |

---

### Author Response · Authors · 2023-08-18
**Presentation Improvements**

We have been improving the presentation of our paper. We have already updated a new version in which we fixed all the errors and typos pointed out by the reviewers and included some new results. Later in this discussion phase, we will post another update which includes more revisions (e.g., new results, new analysis). In this message, we give a briefing describing the revisions.

First, we emphasize our contributions as discussed in [Contribution and Significance].

Second, we rename Sec-3 to Background and include visual examples of event sequences, along with visual illustration of key concepts of "intensity" and "point process" (suggested by R.JyMg). This revision will make the paper more friendly to novices in the field.

Third, we add all the new results, along with relevant discussion; you can see them in [New Results].

Fourth, we add a new section discussing new research opportunities inspired by our results, which you can see in [New Analysis]. This new section will make the paper more "juicy" for experienced researchers in this field.

---

### Author Response · Authors · 2023-08-18
**New Analysis**

R.Q2cp and R.mYWh are concerned with "what's new/novel in this work", and ask for insights about future research opportunities.
We sincerely apologize for not including such analysis in the submitted version: we thought about it, but left it out due to page limit. With the extra page allowed by camera-ready, it would be easy to add it to the final version.

In this rebuttal, we summarize our thoughts on future research opportunities inspired by our benchmarking results. However, we feel eager to emphasize: besides the new insights that we'll outline below, our library itself would significantly facilitate the creation of new methods and the discovery of new insights in this research area; it seems to be what NeurIPS Datasets and Benchmarks track has been calling for, isn't it?

Most importantly, the results across datasets and architectures seem to be signaling that we should think beyond architectural design. For the past decade, this area has been focusing on developing new architectures, but the performance of new models on the standard datasets seem to be saturating. Notably, all the best to-date models make poor predictions on time of future events. Moreover, on type prediction, attention-based models only outperform other architectures by a small margin. Looking into the future, we advocate for a few new research directions that may bring significant contributions to the field.

The first is to build foundation models for event sequence modeling. The previous model-building work all learns data-specific weights, and does not test the transfering capabilities of the learned models. Inspired by the emergence of foundation models in other research areas, we think it will be beneficial to explore the possibility to build foundation models for event sequences. Conceptually, learning from a large corpus of diverse datasets---like how GPTs learn by reading open web text---has great potential to improve the model performance and generalization beyond what could be achieved in the current in-domain in-data learning paradigm. Our library can facilitate exploration in this direction since (as explained in [Contribution and Significance]) we unify the data formats and provide an easy-to-use interface that users can seamlessly plug and play any set of datasets. Challenges in this direction arise because different datasets tend to have disjoint sets of event types and different scales of time units. Ways to tackle such challenges may include:
- sharing some (e.g., attention, transition function) but not all of the model parameters across datasets;
- leveraging more features of each dataset (e.g., text description of event types, meta info of data);
- exploring hierarchical architectures for the foundation model to handle varying time scales (e.g., minutes, days, years)

The second is to go beyond event data itself and utilize external information sources to enhance event sequence modeling. Seeing the performance saturation of the models, we are inspired to think whether the performance has been bounded by the intrinsic signal-to-noise ratio of the event sequence data. Therefore, it seems natural and beneficial to explore the utilization of other information sources, which include but are not limited to:
- sensor data such as satellite images and radiosondes signals, which may help predict natural hazard events (e.g., tsunami, earthquake);
- structured and unstructured knowledge bases (e.g., databases, Wikipedia, textbooks), which contain rich knowledge about real-world events that could help with prediction
- large pretrained models such as GPTs, whose rich knowledge and strong reasoning capabilities may assist event sequence models in improving their prediction accuracies.

The third is to go beyond observational data and embed event sequence models into real-world interventions. With interventional feedback from the real world, an event sequence model would have the potential to learn real causal dynamics of the world, which may significantly improve prediction accuracy.

All the aforementioned directions open up research opportunities for technical innovations.

---

### Author Response · Authors · 2023-08-18
**Commitment**

We are committed to providing continued and high-quality service to the community. Particularly, we will follow research progress in this area and actively add new ingredients to our library. At the same time, we warmly welcome open contributions (as we have stated in the paper), and hope that we could build a sustainable open-source community for this research area.

The Todo List on our Github page has been tracking the datasets and models that we plan to add to the repository, and it has already listed the kinds of data and models suggested by R.3bs8 and R.JyMg.

---

### Author Response · Authors · 2023-08-18
**Contribution and Significance**

Time-stamped event data is ubiquitous in real-world domains, including finance, healthcare, and social media. Temporal point processes (TTPs) are a natural tool for modeling such data. As shown in Fig-1 of our paper, recent years have seen a thousand new research papers on the topic of event sequence modeling and temporal point processes. In this work, we take the initiative to build a central library of popular research assets (e.g., data, models, eval methods, docs)---inspired by Hugging Face for computer vision and natural language processing---in order to facilitate future research in this area. The community seems to really appreciate this initiative: without any advertising, our library has collected >80 stars on Github and has been downloaded around 700 times from PyPi since it was released (~2 months ago).

Q2cp is concerned that "no new datasets or evaluation tasks" and that all "have been used by previous studies". It is true. But it is also true that a central repository of these existing resources---which is non-trivial to put together---will significantly facilitate future research in this area:
- we unify the data format and provide source code (with thorough documentation) for data processing. It will free future researchers from large amounts of data-processing efforts. The unified interface will also facilitate new research topics in this area; please see [New Analysis] for details.
- we provide a diverse and extendable suite of evaluation methods. The suite contains popular evaluation methods (e.g., log-likelihood, next-event prediction accuracy, long-horizon prediction accuracy), and is ready for future researchers to use off-the-shelf. Our evaluation code from previous papers has been widely used in the community; in this work, we clean and share it with easy-to-use interface. Using a shared suite of evaluation code will not only help researchers speed up their development, but also make their results more reproducible. Moreover, following our documentation and protocols, one could easily implement and publicize their own evaluation methods.

In addition, our library will significantly facilitate future method development because it provides a rich suite of modules (functions and classes): we reproduced previous models by composing these modules like building LEGOs; other researchers can reuse our modules to build their new models, which we believe would significantly accelerate their implementation and improve their experience. Here are 2 examples that will be largely useful:
- compute_loglikelihood (function), which calculates log-likelihood of a model given data. It is non-trivial to correctly implement it due to the integral term of log-likelihood (see eq-4 in paper), and we have found errors in popular implementations.
- EventSampler (class), which draws events from a given point process via the thinning algorithm. The thinning algorithm is commonly used in inference but it is non-trivial to implement (and rare to see) an efficient and batched version. Our efficient and batched version (which we took great efforts to implement) will be useful for nearly all intensity-based event sequence models.

Our library is compatible with both pyTorch and TensorFlow, the top-2 popular deep learning frameworks, and thus offers a great flexibility for future research in method development.

Furthermore, following our documentation, one could easily design and implement their own modules, by either composing our existing modules or writing pyTorch/TensorFlow code from scratch.

---

### Author Response · Authors · 2023-08-18
**To-all Message**

We thank reviewers for constructive feedback! We will kickstart our response with a few to-all messages, clarifying our contributions and innovations, presenting new results and new discussion, as well as proposing improvements to the paper. We will address other concerns in messages to individual reviews.

Our to-all messages are organized as follows:
- [Contribution and Significance]: We emphasize the contribution of this work to the community.
- [Commitment]: We highlight our commitment to maintaining this library, and discuss the data and models mentioned by each reviewer.
- [New Analysis]: We address concerns about "what's new in your work" raised by R.Q2cp and R.mYWh.
- [Presentation Improvements]: We discuss the presentation improvements that we will execute for the camera-ready version.
- [Sanity Check]: We present our sanity check results for our reimplementation of the models.
- [New Results]: We present results of the experiments requested by each reviewer.

---

### Author Response · Authors · 2023-08-22
**Paper Updated (updated at 08-22)**

Dear reviewers,

We have uploaded a new version, incorporating the edits proposed in [Presentation Improvements].

The new content (e.g., text, figures, tables) is highlighted in light-green background color (which we will turn off for camera-ready). Note that, due to the new content, figures and tables have been re-indexed and thus are different from the previous version.

Particularly, the green-colored new content includes:
- a clarification of our contributions in Introduction, as discussed in [Contribution and Significance].
- a renamed and updated Sec-3 (Background) with a new figure (Fig-2) illustrating key technical concepts of "intensity" and "point process". We hope this new version is more friendly and accessible to beginners in this field.
- new results and analysis (Sec-6.2) presented in the rebuttal (e.g., MHP, FullyNN, IFTPP). In addition, we have done: for a clearer comparison, we show actual numbers of elements of Fig-6 to 8 in Tab-5 to 6 in Appendix D.3; we present Torch vs. TensorFlow in Tab-2 rather than in figures; sanity check results are added in Appendix D.2.
- a new section Sec-7 (Future Research Opportunities) discussing new research directions inspired by our results.

We hope that this new version can set your minds at ease. Should you have any further advice on the paper and/or our rebuttal, please let us know and we will be more than happy to engage in more discussion and paper improvements!

---

### Decision · Program_Chairs · 2023-09-22

**Decision:**

Reject

**Comment:**

The paper conducts an empirical examination of several recent temporal point process models. It is an important topic and very much needed for the community. At the same time, rigorous work is needed given its importance. There is still room for improvement in this version of the submission. For example, how do the results reported in the paper match the existing literature? what leads to improvement in some methods but files in other methods? In general, a good survey paper not only involves comparing results but also (and most importantly) the insights through this examination. The paper could benefit from one more round of revision before being ready for publication.